# Decomposing NeRF for Editing
# via Feature Field Distillation

**Sosuke Kobayashi**
Preferred Networks, Inc.
`sosk@preferred.jp`

**Eiichi Matsumoto**
Preferred Networks, Inc.
`matsumoto@preferred.jp`

**Vincent Sitzmann**
Massachusetts Institute of Technology
`sitzmann@mit.edu`

`pfnet-research.github.io/distilled-feature-fields/`

## Abstract

Emerging neural radiance fields (NeRF) are a promising scene representation for computer graphics, enabling high-quality 3D reconstruction and novel view synthesis from image observations. However, editing a scene represented by a NeRF is challenging, as the underlying connectionist representations such as MLPs or voxel grids are not object-centric or compositional. In particular, it has been difficult to selectively edit specific regions or objects. In this work, we tackle the problem of semantic scene decomposition of NeRFs to enable query-based local editing of the represented 3D scenes. We propose to distill the knowledge of off-the-shelf, supervised and self-supervised 2D image feature extractors such as CLIP-LSeg or DINO into a 3D feature field optimized in parallel to the radiance field. Given a user-specified query of various modalities such as text, an image patch, or a point-and-click selection, 3D feature fields semantically decompose 3D space without the need for re-training and enable us to semantically select and edit regions in the radiance field. Our experiments validate that the distilled feature fields can transfer recent progress in 2D vision and language foundation models to 3D scene representations, enabling convincing 3D segmentation and selective editing of emerging neural graphics representations.

## 1   Introduction

Emerging neural implicit representations or neural fields have been shown to be a promising approach for representing a variety of signals [82, 53, 65, 106, 56]. In particular, they play an important role in 3D scene reconstruction and novel view synthesis from a limited number of context images. Neural radiance fields (NeRF) [56] enabled the recovery of a continuous volume density and radiance field from a limited number of observations, producing high-quality images from arbitrary views via volume rendering with promising applications in computer graphics. However, editing a scene reconstructed by NeRF is non-obvious because the scene is not object-centric and is implicitly encoded in the weights of a connectionist representation such as an MLP [56] or a voxelgrid [23]. Although we can transform the scene in input or output space or via optimization-based editing [37, 97], this does not enable selective *object-centric* or *semantic*, local edits, such as moving a single object. Prior work has addressed this challenge via coordinate-level, semantic decompositions which allow to selectively move, deform, paint, or optimize parts of a NeRF, but relies on costly annotation of instance segmentations and training of instance-specific networks [104]. While this can be alleviated with pre-trained segmentation models [25, 41], such models require pre-defined closed label sets and

36th Conference on Neural Information Processing Systems (NeurIPS 2022).

domains (e.g., traffic scenes), limiting decomposition and editing. Local editing of NeRFs ideally requires an efficient, open-set method for coordinate-level decomposition.

In this work, we present *distilled feature fields* (DFFs), a novel approach to query-based scene decomposition for local, interactive editing of NeRFs. We focus on 3D neural *feature* fields, which map every 3D coordinate to a semantic feature descriptor of that coordinate. Conditioned on a user query such as a text or image patch, this 3D feature field can compute a decomposition of a scene without re-training. We train a scene-specific DFF via teacher-student distillation [34], using supervision from feature encoders pre-trained on the image domain. Unlike the domain of 3D scenes, the image domain boasts massive high-quality datasets and abundant prior work on self-supervised and supervised training of effective feature extraction models. Notably, recently proposed transformer-based models [96, 22] have demonstrated impressive capabilities across various vision- and text-based tasks (e.g., CLIP [69], LSeg [44], DINO [12]). Such feature spaces capture the semantic properties of regions and make it possible to correspond and segment them well by text, image queries, or clustering. We employ these models as teacher networks and distill them into 3D feature fields via volume rendering. The trained feature field enables us to semantically select and edit specific regions in 3D NeRF scenes and render multi-view consistent images from the locally edited scenes.

In extensive experiments, we investigate the applications of neural feature fields with two different pre-trained teacher networks, (1) LSeg [44], a CLIP-inspired language-driven semantic segmentation network, and (2) DINO [12, 3], a self-supervised network aware of various object boundaries and correspondences. LSeg and DINO features allow us to select 3D regions by a simple text query or an image patch, respectively. We first quantitatively demonstrate that LSeg-based DFFs with label queries can have high 3D segmentation performance compared with an existing point-cloud based 3D segmentation baseline trained on ScanNet [20], a supervised point-cloud dataset. We then demonstrate a variety of 3D appearance and geometry edits across real-world NeRF scenes with no annotations of segmentation; and show that we may edit regions with a single query of text, image, pixel, or cluster choice.

## 2 Related Work

**Neural Implicit Representations.** Neural implicit representations or neural fields have recently advanced neural processing for 3D data and multi-view 2D images [82, 53, 65, 106, 56]. For a review of this emerging space we point the reader to the reports by Kato et al. [39], Tewari et al. [90], and Xie et al. [102]. In particular, a neural radiance field (NeRF) can be fitted to a set of posed 2D images and maps a 3D point coordinate and a view direction to RGB color and density. When observations are limited, NeRF often overfits and fails to synthesize novel views with correct geometry and appearance. Pre-trained vision models have been used for regularizing NeRF via flows [62], multi-view consistency [35], perceptual loss [110], or depth estimation [100, 77]. Some pre-trained models operate not only in the visual world but also in other modalities such as language. The recently proposed CLIP model [69] has demonstrated impressive performance in image-and-text alignment, with strong generalization to various textual and visual concepts. Wang et al. [97], Jain et al. [36], and Poole et al. [68] use CLIP or Imagen [79] to edit or generate a single-object NeRF with a text prompt query by optimizing the NeRF parameters to generate images matched with the text. While such methods are promising, they do not enable accurate selective editing of specific scene regions. For example, the prompt "yellow flowers" may affect unintended scene regions, such as the leaves of a plant. Our proposed decomposition method leverages pre-trained foundation models to enable selective editing of real-world NeRF scenes. Neural descriptor fields [80] use intermediate features that emerge in a 3D occupancy field network [53] for efficiently teaching robots object grasping. Instead of a pre-trained object-centric 3D model, we use 2D vision models as teacher networks via distillation, exploiting recent progress in pre-trained foundation models [7].

**Geometric Decomposition of Neural Scene Representations** Kohli et al. [40] and Zhi et al. [112] show that neural implicit representations can be combined with the supervision of semantic labels. Yang et al. [104] demonstrate that given view-consistent ground-truth instance segmentation masks during training, NeRF can be trained to represent each object as different volumes, although such an annotation is expensive in practice. Concurrently, Benaim et al. [6] also experiment with the different parametarization. Conditional [49, 37, 21, 63] and generative models [60, 61, 31] enable a degree

of category-specific decomposition (e.g., human bodyparts) and editing on constrained domains with large datasets. Regular structures such as voxelgrids or octrees [13, 48, 14, 88, 94, 89, 43, 81, 107, 59, 60] or unsupervised decomposition [73, 85, 109, 83] enable editability via manipulation of localized parameters. However, the decomposition is limited due to the inflexibly structured boundaries or strong assumptions about scenes; self-supervised object-centric learning is a difficult task. Other studies also explored reconstruction with more structured hybrid representations via pipelines specialized to a domain (e.g., traffic scene) [64, 25, 41] or situation (e.g., each object data is independently accessible) [28, 27, 105]. Note that this line of work defines and constrains domains or the types of segmentation during or before training and thus limits the degrees of freedom for editable scenes and objects. In contrast, our method can decompose scene-specific NeRFs into arbitrary semantic units via text and image queries, enabling versatile scene edits without re-training. A concurrent paper by Tschernezki et al. [93] also explores the same training framework and, in particular, investigates how fused features are improved from 2D teacher networks. It also complementarily shows the results with other teacher models (MoCo-v3 [17] and DeiT [91]), dimension reduction via PCA, and NeuralDiff [92]-based neural fields. Other concurrent studies explore decomposition through training scene-specific segmentation field [113] or 3DCNN [76] supervised by click or scribble annotations. Lastly, in a different but related task, video editing, Kasten et al. [38] use foreground-background decomposition and atlas representation for time-consistent, local editing; Loeschcke et al. [51] and Bar-Tal et al. [5] further use CLIP for editing.

**Zero-shot Semantic Segmentation.** Zero-shot semantic segmentation is a challenging task [24, 2, 10] where a model has to predict semantic labels of pixels in images without a-priori information of the categories. A typical solution is to use vision-and-language cross-modal encoders. They are trained to encode images (pixels) and text labels into the same semantic space and perform zero-shot prediction based on the similarity or alignments of the two inputs. Recent development of image encoder architectures [96, 22, 71] and large-scale training [69, 12] have improved the ability and generalization of vision models, including zero-shot models [44, 52, 99, 103, 114, 72]. On the other hand, ongoing studies on zero-shot perception in 3D still suffer from the lack of effective, efficient, and high-resolution architectures and large-scale annotated datasets [54, 33, 29, 101, 78, 30]. Our method is a new approach to perform zero-shot semantic segmentation on scene-specific 3D fields by exploiting progress in the image domain without semantic 3D supervision. We note that the goal of this paper is not to achieve state-of-the-art performance on 3D semantic segmentation tasks. Instead, our goal is the decomposition of neural scene representations for editing, which requires smooth segmentation results on continuous 3D space rather than segmentation of discrete point clouds or voxelgrids.

## 3 Preliminaries

### 3.1 Neural Radiance Fields (NeRF)

NeRF [56] uses MLPs to output density $\sigma$ and color $\mathbf{c}$ given a point coordinate $\mathbf{x} = (x, y, z)$ in a 3D scene. This simple scene representation can be rendered and optimized via volume rendering. Given a pixel's camera ray $\mathbf{r}(t) = \mathbf{o} + t\mathbf{d}$, depth $t$ with bounds $[t_{\text{near}}, t_{\text{far}}]$, camera position $\mathbf{o}$, and its view direction $\mathbf{d}$, NeRF calculates the color of a ray using quadrature of $K$ sampled points $\{\mathbf{x}_k\}_{k=1}^{K}$ with depths $\{t_k\}_{k=1}^{K}$ as

$$\hat{\mathbf{C}}(\mathbf{r}) = \sum_{k=1}^{K} \hat{T}(t_k)\, \alpha\left(\sigma(\mathbf{x}_k)\delta_k\right) \mathbf{c}(\mathbf{x}_k, \mathbf{d}), \quad \hat{T}(t_k) = \exp\left(-\sum_{k'=1}^{k-1} \sigma(\mathbf{x}_{k'})\delta_{k'}\right), \quad (1)$$

where $\alpha(x) = 1 - \exp(-x)$, and $\delta_k = t_{k+1} - t_k$ is the distance between adjacent point samples. NeRFs are optimized solely on a dataset of images and their camera poses by minimizing a re-rendering loss.

### 3.2 Pre-trained Models and Zero-shot Segmentation of Images

Most semantic segmentation models pre-define a closed set of labels, and cannot flexibly change the segmentation categories or boundaries without supervised training. In contrast, zero-shot semantic segmentation predicts target regions given open-set queries. Li et al. [44] proposes LSeg, a model to

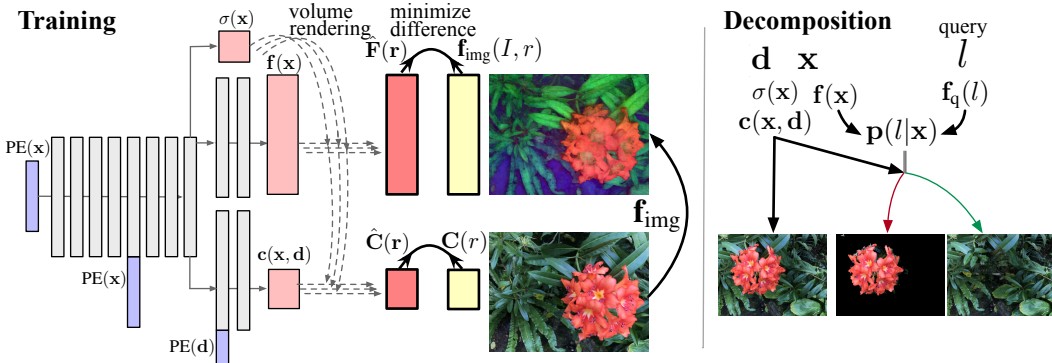

Figure 1: Left: A Distilled Feature Field (DFF) maps a coordinate $\mathbf{x}$ and a viewing direction $\mathbf{d}$ to density $\sigma$, color $\mathbf{c}$, and feature $\mathbf{f}$. It is trained by minimizing the difference between rendered features and features as predicted by a pre-trained image feature encoder, as well as the rendered color and ground-truth pixel color. Right: At test time, we may decompose and edit 3D space via selecting and manipulating different 3D regions with a variety of queries.

perform zero-shot semantic segmentation by aligning pixel-level features and a text query feature. LSeg employs an image feature encoder with the DPT architecture [71] and a CLIP-based text label feature encoder [69], trained via large-scale language-image contrastive learning. The probability of a text label $l$ given a pixel $r$ in an image $I$, $\mathbf{p}(l|I,r)$, is then calculated via dot product of pixel-level image feature $\mathbf{f}_{\text{img}}(I,r)$ and queried text feature $\mathbf{f}_{\text{q}}(l)$ followed by a softmax:

$$\mathbf{p}(l|I,r) = \frac{\exp(\mathbf{f}_{\text{img}}(I,r)\mathbf{f}_{\text{q}}(l)^{\text{T}})}{\sum_{l' \in \mathcal{L}} \exp(\mathbf{f}_{\text{img}}(I,r)\mathbf{f}_{\text{q}}(l')^{\text{T}})} \ , \tag{2}$$

where $\mathcal{L}$ is a set of possible labels. If negative labels are not available, we may use other scores like thresholded cosine similarity to directly compute the probability of a label. During training, LSeg optimizes only the image encoder $\mathbf{f}_{\text{img}}(I,r)$ by minimizing cross-entropy on supervised semantic segmentation datasets. The text encoder $\mathbf{f}_{\text{q}}(l)$ is obtained from a pre-trained CLIP model [69]. Recently, pre-trained CLIP has been leveraged as the backbone for a variety of tasks and has been extended with additional modules sharing the same latent space. For example, Reimers and Gurevych [74, 75] trains a multi-lingual (more than 50+ languages) text encoder, which enables CLIP and CLIP-inspired variants to use non-English queries like Japanese. We similarly use the latent space of a pre-trained CLIP for LSeg via distillation, enabling the decomposition of NeRFs with both English and non-English queries. Segmentation can further be performed with other modalities such as image, patch or pixel query features $\mathbf{f}_{\text{q}}$ using a similar dot-product similarity formulation as in Eq. 2. Notably, DINO [12], a self-supervised vision model, solves video instance segmentation and tracking by calculating similarity among features in adjacent frames. Amir et al. [3] also demonstrate that DINO features work well on co-segmentation and point correspondence by similarity and clustering. In our experiments, we use these two publicly available models, LSeg and DINO, to obtain features of images and texts for 3D decomposition.

## 4 Distilled Feature Fields

### 4.1 Distilling Foundation Modules into 3D Feature Fields via Volume Rendering

NeRF learns a neural field to compute the density and view-dependent color, $\sigma(\mathbf{x})$ and $\mathbf{c}(\mathbf{x}, \mathbf{d})$. We may extend NeRF by adding decoders for other quantities of interest. For example, SemanticN-eRF [112] adds a branch outputting a probability distribution of closed-set semantic labels, trained with supervision via images with ground-truth semantic labels. This enables the prediction of pairs of RGB and semantic segmentation masks from novel views, useful for data augmentation. However, because ground-truth annotation is costly, the method is inefficient as a means of scene editing [104]. For specific domains like traffic scenes [25, 41], we may instead train a closed-set segmentation model and use its prediction for training object-aware neural fields. However, this approach is possible only if the types of objects are limited and the domain-specific supervised dataset is available; limiting the application of scene editing in terms of domain and flexibility of decomposition.

We build on top of these ideas and perform 3D zero-shot segmentation of NeRFs using open-set text labels or other feature queries. Instead of a branch performing closed-set classification, we propose to add a feature branch outputting a feature vector itself. This branch models a 3D feature field describing the semantics of each spatial point. We supervise the feature field by a pretrained pixel-level image encoder $\mathbf{f}_{\text{img}}$ as a teacher network. Given a 3D coordinate $\mathbf{x}$, the feature field outputs a feature vector $\mathbf{f}(\mathbf{x})$ in addition to density $\sigma(\mathbf{x})$ and color $\mathbf{c}(\mathbf{x}, \mathbf{d})$, as shown in Fig. 1. Volume rendering of the feature field is similarly performed via

$$\hat{\mathbf{F}}(\mathbf{r}) = \sum_{k=1}^{K} \hat{T}(t_k)\, \alpha(\sigma(\mathbf{x}_k)\delta_k)\, \mathbf{f}(\mathbf{x}_k) \ . \tag{3}$$

We can optimize $\mathbf{f}$ by minimizing the difference between rendered features $\hat{\mathbf{F}}(\mathbf{r})$ and the teacher's features $\mathbf{f}_{\text{img}}(I, r)$. Effectively, we are distilling [34] the 2D teacher network into our 3D student network via differentiable rendering, and thus dub this model a *distilled feature field* (DFF). We add a feature objective $\mathcal{L}_f$ penalizing the difference between rendered features $\hat{\mathbf{F}}(\mathbf{r})$ and the teacher's outputs $\mathbf{f}_{\text{img}}(I, r)$ to the photometric loss of the original NeRF. We use two networks for volume rendering with coarse-and-fine hierarchical sampling. We thus minimize the sum of photometric loss $L_p$ and feature loss $L_f$, in total, $L$:

$$L = L_p + \lambda L_f, \quad L_p = \sum_{\mathbf{r} \in \mathcal{R}} \left\| \hat{\mathbf{C}}(\mathbf{r}) - \mathbf{C}(r) \right\|_2^2, \quad L_f = \sum_{\mathbf{r} \in \mathcal{R}} \left\| \hat{\mathbf{F}}(\mathbf{r}) - \mathbf{f}_{\text{img}}(I, r) \right\|_1, \tag{4}$$

where $\mathcal{R}$ are sampled rays, $\mathbf{C}(r)$ is the ground truth pixel color of ray $r$, $\lambda$ is the weight of the feature loss and is set to 0.04 to balance the losses [112]. We apply stop-gradient to density in rendering of features $\hat{\mathbf{F}}(\mathbf{r})$ in Equation 3 as the teacher's features $\mathbf{f}_{\text{img}}(I, r)$ are not fully multi-view consistent, which could harm the quality of reconstructed geometry.

## 4.2 Query-based Decomposition and Editing

A trained DFF model can perform 3D zero-shot segmentation by its feature field $\mathbf{f}$ and a query encoder $\mathbf{f}_{\text{q}}$. Probability of a label $l$ of a point $\mathbf{x}$ in the 3D space, $\mathbf{p}(l|\mathbf{x})$, is calculated by dot product of the 3D feature $\mathbf{f}(\mathbf{x})$ and text label feature $\mathbf{f}_{\text{q}}(l)$ followed by a softmax:

$$\mathbf{p}(l|\mathbf{x}) = \frac{\exp(\mathbf{f}(\mathbf{x})\mathbf{f}_{\text{q}}(l)^{\text{T}})}{\sum_{l' \in \mathcal{L}} \exp(\mathbf{f}(\mathbf{x})\mathbf{f}_{\text{q}}(l')^{\text{T}})} \ . \tag{5}$$

This query-based segmentation field is at the core of the proposed method. It can be calculated at any 3D point without limiting resolution, naturally used in tandem with a radiance field and volume rendering. Note that the segmentation depends on only the 3D coordinate and the query[1]. As the original NeRF, it is thus multi-view consistent. In addition and important for interactive editing, we can change the segmentation via queries without re-training, which cannot be realized by closed-set methods using semantic [112] or instance segmentation annotation [104]. We may now use this query-conditional segmentation to identify a specific 3D region for editing. Various edits can be generalized to the merging of two NeRF scenes $\sigma_1(\mathbf{x}), \mathbf{c}_1(\mathbf{x}, \mathbf{d})$ and $\sigma_2(\mathbf{x}), \mathbf{c}_2(\mathbf{x}, \mathbf{d})$, where we use the segmentation field $\mathbf{p}$ for blending. In the experiments section, we simply modify Eq. 1 as a blend of two scenes based on the ratio of $\alpha$:

$$\hat{\mathbf{C}}(\mathbf{r}) = \sum_{k=1}^{K} \hat{T}(t_k) \left( \alpha(\sigma_1(\mathbf{x}_k)\delta_k)\, \mathbf{c}_1(\mathbf{x}_k, \mathbf{d})\rho_k + \alpha(\sigma_2(\mathbf{x}_k)\delta_k)\, \mathbf{c}_2(\mathbf{x}_k, \mathbf{d})(1 - \rho_k) \right) , \tag{6}$$

$$\text{where} \quad \rho_k = \frac{\alpha(\sigma_1(\mathbf{x}_k)\delta_k)}{\alpha(\sigma_1(\mathbf{x}_k)\delta_k) + \alpha(\sigma_2(\mathbf{x}_k)\delta_k)} , \quad \hat{T}(t_k) = \prod_{k'=1}^{k-1} \alpha(\sigma_1(\mathbf{x}_{k'})\delta_{k'}) + \alpha(\sigma_2(\mathbf{x}_{k'})\delta_{k'}) . \tag{7}$$

For example, if we want to apply a geometric transformation $\mathbf{g}$ to a region of a query $l$ in a NeRF scene $(\sigma, \mathbf{c})$, we can render the transformed scene via Eqs. 6 and 7 by setting $\alpha(\sigma_1(\mathbf{x}_k)\delta_k) = (1 - \mathbf{p}(l|\mathbf{x}_k))\alpha(\sigma(\mathbf{x}_k)\delta_k)$, $\alpha(\sigma_2(\mathbf{x}_k)\delta_k) = \mathbf{p}(l|\mathbf{g}^{-1}(\mathbf{x}_k))\alpha(\sigma(\mathbf{g}^{-1}(\mathbf{x}_k))\delta_k)$, $\mathbf{c}_1(\mathbf{x}_k, \mathbf{d}) = (1 - \mathbf{p}(l|\mathbf{x}_k))\mathbf{c}(\mathbf{x}_k, \mathbf{d})$, and $\mathbf{c}_2(\mathbf{x}_k, \mathbf{d}) = \mathbf{p}(l|\mathbf{g}^{-1}(\mathbf{x}_k))\mathbf{c}(\mathbf{g}^{-1}(\mathbf{x}_k), \mathbf{g}^{-1}(\mathbf{d}))$. More details of editing for

---

[1] It is an interesting extension to introduce a user's viewpoint to the function for recognizing view-dependent queries like referring expressions (e.g., "the chair left to the table") [16, 1, 47, 4]. We leave this to future work.

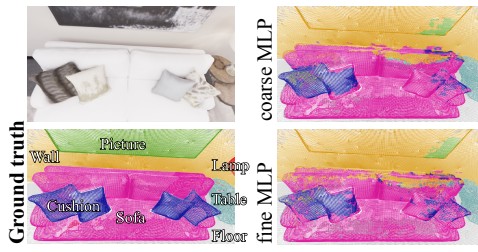

Figure 2: Comparison of predictions by coarse and fine MLPs.

Table 1: Performance of 3D semantic segmentation on Replica dataset. DFF outperforms a *supervised* point-cloud segmentation model MinkowskiNet42.

|  | mIoU | accuracy |
|---|---|---|
| Supervised 3DCNN | 0.475 | 0.758 |
| DFF (Coarse) | 0.589 | 0.855 |
| DFF (Fine) | 0.583 | 0.855 |

colorization, translation, and deletion are shown in Appendix B. We can combine this with more complex edits, including optimization-based methods like CLIPNeRF [97]. While CLIPNeRF itself cannot selectively edit specific regions in multi-object scenes, our decomposition method enables it to update only desired objects without breaking unintended areas.

## 5 Experiments

We first conduct a quantitative evaluation of the decomposition achieved by DFF. We demonstrate that DFF enables 3D semantic segmentation in a benchmark dataset using scanned point clouds with human-annotated semantic segmentation labels. We then investigate the capabilities of DFF for editing and subsequent novel-view synthesis on real-world datasets. We use two teacher networks, LSeg [44] and DINO [12], which are pre-trained and publicly available. Each training image is encoded by the image encoders of the networks and used as target feature maps, $\mathbf{f}_{\mathrm{img}}(I, r)$, defined in Equation 4. Because the feature maps are of reduced sizes due to the limitation of the networks, we first resize them to the original image size. The implementation and settings of NeRF, unless otherwise stated, follow Zhi et al. [112]. During the training of 200K iterations, the loss $L$ in Equation 4 is minimized by Adam with a linearly decaying learning rate (5e-4 to 8e-5). During training, Gaussian noise for density is also applied. The number of coarse and fine samplings is 64 and 128, respectively. The MLP of the neural radiance field consists of eight ReLU layers with 256 dimensions, followed by a linear layer for density, three layers for color, and three layers for feature, as shown in Fig. 1. Positional encoding of length 10 is used for the input coordinate and its skip connection, and that of length 4 is for viewing direction. If an independent MLP is prepared for the feature field, it consists of four layers (with a skip connection at the third layer if the positional encoding is used). The size of a training image is $320 \times 240$ for the Replica dataset and $1008 \times 756$ for the other datasets. The batchsize of training rays is 1024 for Replica and 2048 for the others. During finetuning of feature fields or radiance fields, Gaussian noise is removed, and the learning rate is set to 1e-4. See appendix A and C for further training details.

### 5.1 3D Semantic Segmentation

We construct a 3D semantic segmentation benchmark from four scenes in the Replica dataset [86] with data split and posed images provided by [112]. See appendix D for further details of the dataset. We train DFF to reconstruct each scene with radiance and feature fields from training images and evaluate the quality of novel view synthesis and 3D segmentation of the annotated point clouds. We use LSeg as a teacher network. The LSeg text encoder encodes each label, and the probability of each point is calculated by Equation 2[2]. Note that the training uses only the photometric and feature losses (Equation 4) and does not access any supervision via semantic labels.

**Semantic Segmentation Results.** First, we show evaluation metrics of 3D semantic segmentation, mean intersection-over-union (mIoU) and accuracy in Table 1. For comparison, we also experiment with a sparse 3D convolution-based segmentation model, MinkowskiNet42 [18] taking a colored

---

[2]While LSeg-DFF can perform zero-shot inference using text labels that are not seen during training, we do not focus on thoroughly evaluating the zero-shot ability. The evaluation has been conducted in the original paper on the teacher network, and DFF's ability is expected to follow it due to distillation. Please refer to Li et al. [44] for the detail of the zero-shot ability of LSeg.

Table 2: Performance of novel view synthesis on Replica dataset. PSNR, SSIM, and LPIPS are metrics of image synthesis. $\delta < 1.25$ and absrel are metrics of geometry (depth estimation).

|  | PSNR↑ | SSIM↑ | LPIPS↓ | $\delta < 1.25$↑ | absrel↓ |
|---|---|---|---|---|---|
| basic NeRF | 32.87 | 0.934 | 0.148 | 0.993 | 0.018 |
| DFF | 32.85 | 0.932 | 0.150 | 0.993 | 0.017 |
| DFF (overweighting $\lambda$) | 32.68 | 0.927 | 0.162 | 0.993 | 0.018 |

point cloud as input. It has a standard state-of-the-art architecture for point cloud segmentation and is trained on the ScanNet dataset [20], the largest annotated training dataset of 3D semantic segmentation[3]. Results demonstrate that DFF, taught by LSeg, achieves promising performance, even better than the supervised model. This indicates that DFF succeeds at distilling 3D semantic segmentation from the 2D teacher network.

**Impact of Sampling on Semantic Segmentation.** NeRF employs two MLPs for hierarchical sampling, where the coarse MLP performs volume rendering with fewer points (64) using stratified sampling, and the fine MLP works with importance sampling (192 in total). So, we have two sampling options to train a feature field. Although fine sampling is critical for training accurate radiance fields, segmentation is of significantly lower spatial frequency than texture. We thus analyze the impact of coarse and fine training in Fig. 2. As expected, the coarse model produces smooth segmentations, while the fine version introduces high-frequency artifacts. This smoothness property is important for natural editable novel view synthesis and is discussed again later.

**Compatibility with View Synthesis.** We also check and compare the quality of novel view synthesis with NeRF, which does not learn feature fields. Because the feature branch partially shares the layers with the radiance field (as shown in Fig. 1), learning feature fields could possibly harm the radiance field. Despite this concern, as shown in Tab. 2, the performance of view synthesis is not degraded. Thus, we can train and use the branch-based DFF with small computational and parameter overhead compared to the original NeRF. If we excessively increased the weight of the feature loss, $\lambda \times 10$, it hurt view synthesis while not improving segmentation performance further. We further confirm that training independent, light-weight feature-field MLP, instead of a branch of the radiance-field MLP, achieves semantic segmentation results competitive with the branch-based approach (see appendix Tab. 3 for the result of all variants)[4]. This option is useful especially when we want to introduce DFF decomposition into arbitrary 3D scene representations, including off-the-shelf NeRF models, dynamic NeRFs [26, 66, 45], or meshes, without re-training of the radiance field.

## 5.2 Editable Novel View Synthesis

In the previous section, we quantitatively validated the ability of DFF to perform semantic decomposition. We now discuss the capability for editable view synthesis on real-world scenes, including the LLFF dataset [55] and our own dataset. Our method can be used even for LLFF scenes based on normalized device coordinates. Please see the supplemental web page for further results, including videos. In addition to LSeg using a text query, we also experiment with self-supervised DINO [12] as another teacher network to enable query-based decomposition using image patch queries. Here, we use thresholded cosine similarity to directly compute the probability of a query instead of softmax with negative queries in Eq. 5 and set $\mathbf{p} = 1$ if the similarity exceeds the threshold [5], and $\mathbf{p} = 0$ otherwise for hard decomposition. We first train NeRFs without a feature branch for each scene for 200K iterations ($L_p$) and then finetune them with a feature branch via distillation for 5K iterations ($L_p + \lambda L_f$) since we found that the feature loss converged significantly faster than the photometric loss and short training was thus sufficient. We use coarse sampling for training feature branches and use it for edited rendering with fine sampling. See appendix A for the details.

---

[3]For a fair comparison, the label set follows the ScanNet dataset. We also manually tune the range and scale of input point clouds to maximize the performance of MinkowskiNet42 on the Replica dataset.

[4]Note that, even if we are interested in the feature-field MLP, it is important to simultaneously prepare the radiance-field MLP for reconstructing geometry well and training the feature field via geometrically plausible volume rendering.

[5]The results of the LLFF room scene used a label set, {whiteboard, ceiling, light, television, wall, bin, table, cabinet, cable, chair, box, floor}, for avoiding tuning of thresholds.

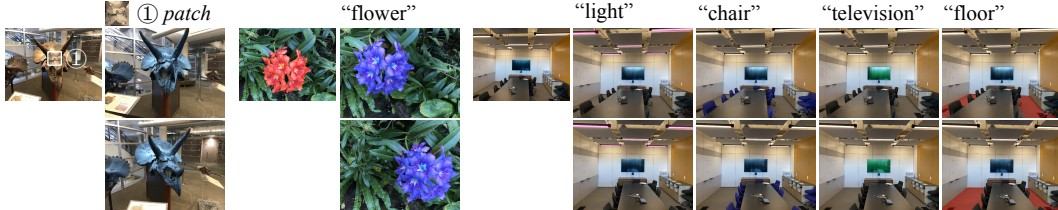

Figure 3: Appearance edits of specific objects via different query modalities: an image patch or text.

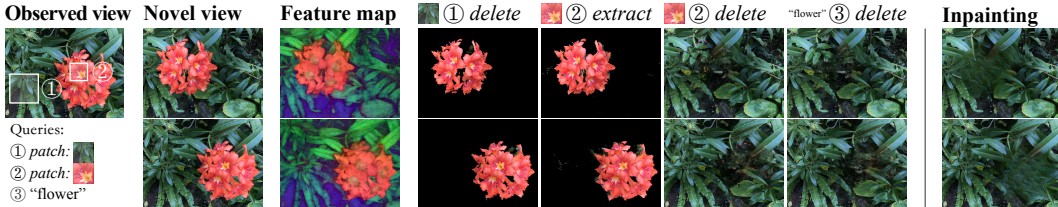

Figure 4: Extraction and deletion of specific objects via different query modalities, an image patch or text. The edited views are 3D consistent, unlike an image inpainting baseline [87]

**Appearance Editing, Deletion, Extraction.** We show qualitative evaluations of novel view synthesis in Fig. 3 and Fig. 4. Specific 3D regions in these scenes are identified and locally edited via decomposition depending on various query modalities. In these experiments, we use a text query for LSeg-DFF as in Section 5.1 and use an image patch query for DINO-DFF. Because DINO features capture the similarity and correspondences of regions well thanks to self-supervised learning [12, 3], image patch queries help select all semantically similar areas at once. The patch feature is then calculated by averaging the features of all pixels in the patch.

In Fig. 3, we demonstrate that the DFF enables convincing selective appearance edits. Because our focus is region selection via decomposition, we use simple color transformation for clarity here (e.g., flip RGB to BGR, blend colors). One might think that the MLP of a radiance field by the original NeRF also has hidden layers, and their features could possibly be used for decomposition. We confirm that the naive usage of NeRF features is not robust to decomposition, as shown in Fig. 5, especially in a complex multi-object scene. We use the 8th hidden layer of the fine radiance field network (i.e., the layer just before branching in Fig. 1)[6]. NeRF features cannot clearly decompose even objects with simple shapes and colors. The region selections are leaked to other parts with similar colors, geometry, or positions while they do not entirely cover the targets. For example, floor selection is leaked to walls, a table, bins, or ceilings. Chair selection is leaked to irrelevant black parts like television, cables, lighting equipment, or shadows. This indicates that the feature space of the original NeRF does not learn semantic similarity well and is entangled with unpredictable and more low-level factors like color or spatial adjacency.

In Fig. 4, we demonstrate that the DFF also works well on deletion or extraction of objects, using two patch queries (query-① for leaves and ground, query-② for flowers) and a text query-③ "flower". For comparison with a baseline editing method, we show the results by a state-of-the-art image inpainting model, LaMa [87]. Because the model requires masks for inpainting regions, we manually annotate the views for evaluation. As shown in the figure, the image inpainting model cannot generate clear and realistic images, and the different views are inconsistent. On the other hand, DFF produces multi-view consistent plausible results, especially succeeding at extracting foreground objects. Although the performance on deleting foreground objects is high, a remaining shortcoming is the existence of floating artifacts and blurred volumes in the far distance behind the deleted object.

**Priors for Smooth Decomposition** We can organize the challenges of editable NeRFs into several categories: surface decomposition, volume decomposition, lighting decomposition, and estimation of less or never observed parts. If we edit appearances only, it practically requires decomposing regions only near the surface of objects, i.e., surface decomposition, because the color of a ray is determined mostly in a condensed interval around the surface. On the other hand, geometric transformations often

---

[6]Other layers or the coarse MLP of the NeRF also indicated similar behaviors but a little worse qualitatively.

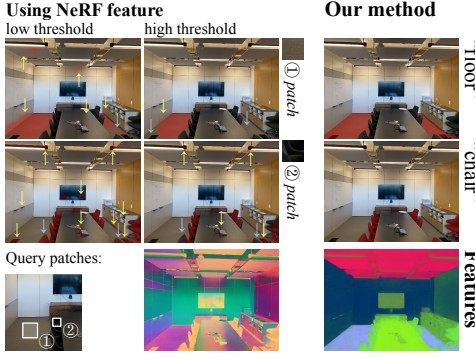

Figure 5: Appearance edits of specific objects, compared with decomposition using features of a NeRF hidden layer. For reference, we also show PCA-based visualizations of the features.

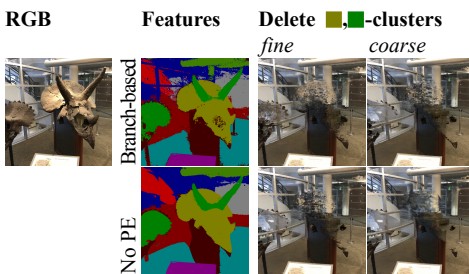

Figure 6: Comparison of predictions by a branch-based feature field MLP and independent MLP with no positional encoding, each of which is trained with coarse and fine sampling.

require a higher level of decomposition. As shown in the deletion examples, geometric transformation may move or remove some surfaces and expose the space behind them. This forces models to render unknown regions less or never observed due to occlusions, including even the inside of objects. Thus, it is desirable to decompose volumes smoothly while synthesizing their inside and back[7]. Although these include the same challenges as novel view synthesis tackles, editability further highlights their importance.

Apart from lighting decomposition discussed in prior work [8, 9, 111], we further investigate the new challenge of smooth volume decomposition by experimenting with different DFF setups. As discussed in Section 5.1, DFF has two sampling options to train feature fields. The coarse training may introduce smoothness regularization and help cohesive decomposition and smoother in-painting of unobserved regions. Another reasonable smoothness regularizer is to eliminate the high-frequency positional encoding (PE). We thus train an independent MLP network for a feature field without PE. We compare four combinations of renderings in Fig. 6. To better understand

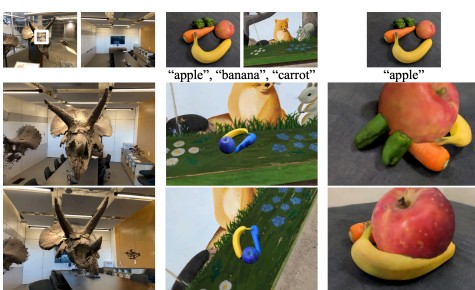

Figure 7: Editing with warping, deformation, shift, and rotation.

their behavior, we use the DINO-DFF, show k-means clusters of the rendered feature map, and delete the head of the Triceratops by a query choosing its corresponding clusters. As expected, coarsely trained models and no-PE models succeed in smoother volume decomposition, and this combination can minimize high-frequency floating artifacts. A side effect is the lack of high-frequency representation power, which sometimes deletes disparate background regions and misses to represent features of complex structures (e.g., see the cluster visualization of the thin frames of the window). Towards the best of both worlds, developing proper priors or inductive biases is an important direction for future work [70]. Otherwise, surface-aware representations like IDR [106, 98] could avoid problems with floating artifacts. Note that not all geometric edits suffer from these problems. For example, it is often less problematic to move objects closer to the camera, enlarge them, or warp them to other scenes, as shown in Fig. 7.

**Localizing Optimization-based Editing.** Finally, we show a combination with an optimization-based editing method. CLIPNeRF [97] optimizes the parameters of a radiance field so that its rendered images match with a text prompt via CLIP. While it is mainly designed for a single-object scene of specific categories, it is possible to apply to other real-world NeRFs. However, because it cannot control the scope of editing, a prompt like "white flower" may change the color of unintentional targets like leaves. Our DFF-based decomposition can upgrade such an optimization-based method

---

[7]Note that this problem also arises when Yang et al. [104] used *ground-truth* instance segmentation masks and trained multiple networks, although the authors did not investigate this issue.

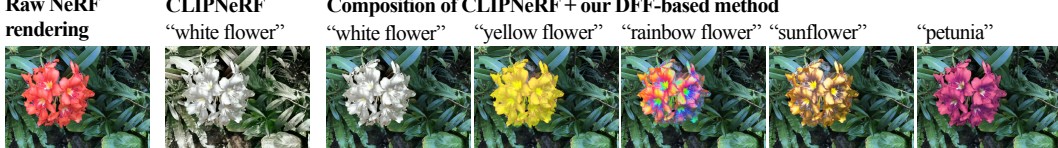

Figure 8: Comparison of appearance editing by CLIPNeRF and our extension.

to render a scene via the composition of a CLIP-optimized NeRF scene and the original NeRF scene. We show the results in Fig. 8[8]. Although the naive CLIPNeRF edits unintentional parts, our method helps it to locally edit intentional parts only. In addition to switching rendering, we can also use the decomposition for controlling training signals during backpropagation. The additional experiment is shown in Appendix F. These extensions broaden the application of CLIPNeRF or other optimization-based editing methods to complex scenes.

## 6 Discussion, Limitations, and Conclusions

In this work, we propose distilled feature field (DFF), a novel method of NeRF scene decomposition for selective editing. We present quantitative evaluations of segmentation and extensive qualitative evaluations of editable novel view synthesis. In addition to these promising results, DFF-based models will benefit from future improvements to self-supervised 2D foundation models. We also clarify future directions on editable view synthesis through our experiments, especially for smoothness priors and estimation of unobserved regions. Furthermore, while this work focuses on editable view synthesis, it is also intriguing to transfer DFF to other applications, including 3D registration of text queries [16, 1, 47, 4] or robot teaching [32, 80].

The limitations of the DFF framework are two-fold. The first one is the upper bduround of the performance due to distillation. The student model of distillation cannot largely outperform the teacher model[9]. If the resolution of teacher encoders is low, the corresponding DFFs also becomes coarse-grained. If the LSeg cannot understand a text query, the LSeg-DFF also cannot. Secondly, the DFF uses volume rendering depending on the 3D reconstruction by NeRF. A NeRF model is sometimes optimized to geometrically wrong solutions (e.g., floaters). Such geometry errors of the radiance fields would make supervision to DFFs noisy.

As a possible negative societal impact, one might use our method for making realistic but fake content by editing NeRFs as desired. Automatic fake detection methods may help in preventing such misuse. NeRFs are further computation-intense, leading to high electricity usage. Recent work on efficient NeRFs [23, 57, 15] may alleviate this concern.

## Acknowledgements

We thank Tsukasa Takagi, Toshiki Nakanishi, Hiroharu Kato, and Masaaki Fukuda for their helpful feedback.

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
