# OpenReview forum: "Decomposing NeRF for Editing via Feature Field Distillation"
_NeurIPS.cc/2022/Conference — NeurIPS 2022 Accept_

### Official Review · Reviewer_1TRr · 2022-07-09

**Rating:** 7
**Confidence:** 3
**Soundness:** 4 excellent
**Presentation:** 4 excellent
**Contribution:** 3 good

**Summary:**

This paper presents a simple yet novel solution to achieve dense volumetric semantic segmentation on NeRF, which supports semantics-guided NeRF decomposition and local editing. This decomposition is achieved by adding an additional renderer that renders the volumetric feature field similarly to radiance by distilling from a pre-trained 2D image semantic feature encoder like CLIP-LSeg. A rich set of experiments is presented to show the decomposition quality, generalization capability to different text and even image queries, and local editing (appearance and geometry) results based on the decomposition.

**Questions:**

I think most of the technical questions and evaluations I would like to see are addressed in the paper. Some minor questions I have:
1. The difference between image segmentation and NeRF segmentation is that image pixels are equally weighted but volumetric sample points have naturally-defined weights (density). Would this cause uncertain/noisy labels in less certain regions (empty space or inside the objects)? I know these regions might not be as important as object surfaces, but just being curious.
2. Is the algorithm robust against artifacts that are commonly seen in real-world NeRF (like misalignment and cloudy artifacts) due to less-perfect camera or input image artifacts?
3. The distillation supervision is applied on alpha-blended renderings, would this be a potential issue that the training tries to overfit the 2D feature supervision with bad volumetric features on the samples? Especially on NeRF trained on narrow baselines? The radiance may have a similar issue but the difference is that the ultimate results for radiance are still rendered views, but here we want to directly use the volumetric features.
4. Is there any boundary issues with the segmentation? Like incomplete regions or holes/noisy points?

**Limitations:**

There is not enough discussion on the limitations as far as I can find. I would suggest the authors add such discussions in the next version.

**Strengths And Weaknesses:**

++ The major strength of this work is that it allows high-quality volumetric NeRF semantic segmentation by reusing the zero-shot image segmentation models.

++ The method is simple and easy to implement, allowing for good results on various applications and works on various NeRF scenes without needing extra training data.

++ Solid experiments and promising results, quantitatively and qualitatively.

-- The method does not show any significant technical breakthrough and is heavily based on LSeg, which limits its technical depth. But I do not think it is a major issue.

---

> ### Author Response · Authors · 2022-08-07
> **Further clarifications?**
>
> Dear reviewer 1TRr,
>
> Thank you for your review! Do you have any remaining questions or concerns following our response? Please let us know. We’d be very happy to do anything we can that would be helpful in the time remaining!

---

> > ### Comment · Reviewer_1TRr · 2022-08-08
> > **Thanks for the response**
> >
> > Thanks for the Authors' response, which I believe clarifies many concerns of mine and other reviewers. Without additional concerns from other reviewers, I would keep my original score and suggest acceptance.

---

### Official Review · Reviewer_cXkV · 2022-07-11

**Rating:** 6
**Confidence:** 4
**Soundness:** 3 good
**Presentation:** 3 good
**Contribution:** 3 good

**Summary:**

The paper studied an important problem, i.e., editing NeRF. It argued that previous works show difficult to selectively edit specific regions or objects. Thus, the paper decomposed scene semantics by learning a semantic feature descriptor distilled from the pretrained CLIP-LSeg or DINO models. The full method DFF (Distilled Feature Field) supports simple query-based decomposition and editing.

**Questions:**

Overall, the presented method is interesting and novel. But there are some unclear points w.r.t. the implementation details and experiments. The reviewer hopes the authors can answer the questions in the "Weaknesses" part.


************************
Thanks for answering my questions. I am happy to raise my rating to "Weak Accept", and hope the authors can carefully addresses the concerns in the revision. Especially, please explain more on the implementation details and clearly present the editing details.

**Limitations:**

Possible limitations have not been discussed.

**Strengths And Weaknesses:**

**Strengths**
1. It is smart to exploit the distilling avenue to allow query-based decomposition and editing for NeRF.

2. The DFF framework is well presented in Figure 1 (left). And the method part clearly explains how to learn a semantic feature descriptor by taking the pretrained CLIP-LSeg or DINO models as teachers.

3. The paper examines different editing operations and shows many qualitative results to demonstrate the effectiveness of DFF.


**Weaknesses**

While the DFF approach is really interesting, the reviewer has some concerns w.r.t. the implementation details and experiments.

1. The reviewer might miss some details, but it seems the paper has not clearly presented how to perform scene editing after training a DFF. Do we need to individually train a target scene for each editing operation?

2. As the paper has stated that the code of complete reproduction of all the results is not yet publicly available, it would be better to include the training details in the main paper.

3. Is it possible to make some quantitate comparisons with other NeRF editing approaches?

---

> ### Author Response · Authors · 2022-08-07
> **Further clarifications?**
>
> Dear reviewer cXkV,
>
> Thank you for your review! Do you have any remaining questions or concerns following our response? Please let us know. We’d be very happy to do anything we can that would be helpful in the time remaining!

---

### Official Review · Reviewer_XqPY · 2022-07-11

**Rating:** 5
**Confidence:** 5
**Soundness:** 2 fair
**Presentation:** 3 good
**Contribution:** 3 good

**Summary:**

To edit a scene more easily, the authors propose a semantic scene decomposition of NeRF by leveraging distillation techniques to distill the 2D feature extractor such as LSeg or DINO into a 3D feature field. Therefore, the distilled NeRF can segment the regions of interest with given texts.

**Questions:**

1. Overall, the paper introduces a distillation method for NeRF semantic segmentation. Although it works well on several scenes, it is highly suggested to test your method on other indoor scenes of LLFF dataset and compared with SOTA indoor 3D semantic segmentation dataset (e.g. Scannet), as you already adopted a highly complicated 2D segmentation model.
2. The authors verify their method based on vanilla NeRF. How about the generalization ability on other NeRF variants? For example, MipNeRF, NeRF for unbounded scenes? You can try to integrate the distillation into an unbounded NeRF variant and show the segmentation quality/view consistency effect by quantitative metrics and visualized videos (e.g. on KITTI dataset with some SOTA segmentation models).

**Ethics Review Area:**

["I don’t know"]

**Limitations:**

See the weakness and questions.

**Strengths And Weaknesses:**

Strengths:
1. The proposed method is easy to understand and follow.
2. The edited results on scene "Flower" looks fancy.

Weaknesses:
1. The claim is not rigorous. The authors claim "feature encoders pre-trained in a self-supervised framework on the image domain". However, this is partly true as LSeg requires semantic segmentation annotation for training.
2. As the author argues "We note that the goal of this paper is not to achieve state-of-the-art performance on 3D semantic segmentation tasks.". However, this will weaken your contribution as you already learned from a large-scale pre-trained model (LSeg), but do not plan to achieve SOTA performance even on NeRF-related datasets. Further, you compare only one 3D segmentation method in Table 1, which contradicts your claim.
3. Mixed-use of Figure and Fig.

---

> ### Author Response · Authors · 2022-08-07
> **Further clarifications**
>
> Dear reviewer XqPY,
>
> Thank you for your review! Do you have any remaining questions or concerns following our response? Please let us know. We’d be very happy to do anything we can that would be helpful in the time remaining!

---

### Official Review · Reviewer_48WN · 2022-07-13

**Rating:** 7
**Confidence:** 4
**Soundness:** 4 excellent
**Presentation:** 4 excellent
**Contribution:** 3 good

**Summary:**

The authors propose a novel approach for getting a NeRF to output both (i) the standard color, and, novel for this work (ii) zero shot-trained features that can be used for semantic segmentation / decomposition / editing. Put differently, whereas previous work had parallel branches for generating, eg, closed-set semantic segmentation labels, this work generates features that can be used with standard zero-shot segmentation techniques. These features are trained via a distillation using LSeg / DINO-based teacher networks.

**Questions:**

Would the approach be able to train just the feature generation branch? While that would not be very useful in the current context, it would nevertheless be interesting to know if the image branch is absolutely necessary for training the feature branch.

**Ethics Review Area:**

["I don’t know"]

**Limitations:**

It would be useful to have a discussion on potential failures.

**Strengths And Weaknesses:**

Positive:
- The paper is generally well written and easy to understand.
- The topic is both very relevant and novel -- there is much work on the geometric side of NeRF, but much less on semantics and even less on feature generation and semantics / editing.
- The approach is principled.

Negative:
- While the feature generation parts is fairly well explored, I found the actual editing part to not be explained very much /  very well. Unless I'm missing something, most of the text focuses on the generation of the features and the decomposition, but the editing is captured only by "We can combine this with more complex edits, including optimization-based methods like CLIPNeRF". I would very want more detail to be added here.
- While the overall structure of the proposed approach is novel (to my knowledge), the individual components are not. I would almost go as far as to say that I see this as a systems paper.

Overall I am quite positive about the paper -- (i) the methodology, while simple, is clear and appears novel; (ii) the results are very good. If accepted, I think the work might end up being quite influential.

---

> ### Author Response · Authors · 2022-08-07
> **Further clarifications?**
>
> Dear reviewer 48WN,
>
> Thank you for your review! Do you have any remaining questions or concerns following our response? Please let us know. We’d be very happy to do anything we can that would be helpful in the time remaining!

---

> > ### Comment · Reviewer_48WN · 2022-08-10
> > **Thank you for the response**
> >
> > Thank you for the response. I am happy to keep my original score and suggest acceptance.

---

### Author Response · Authors · 2022-08-02
**Author Response - General Clarifications**

We thank the reviewers for their careful reading, and detailed and considerate feedback.
We are glad that reviewers think that our method is “both very relevant and novel” with a “principled approach" (48WN), “studies an important problem” and is “well presented” (cXkV), and shows “solid experiments and promising results, quantitatively and qualitatively” (1TRr).
The reviewers also agree that additional experiment & training details, as well as a better discussion of failure cases and limitations, will serve to make the paper stronger, and better inspire future work. We agree. Below, we offer several clarifications and discuss how we will integrate reviewer feedback.

## Details on Editing using Feature Fields
Reviewers remarked a lack of details especially on how exactly the editing was performed (48WN). We were indeed constrained by the page limit, and thus opted to provide only a general formulation of decomposition-and-editing with Equations (5-7) in Section 4.2. However, we agree that more editing details will benefit the paper and will include them in a new Section “4.3 Editing Operations”.

Editing for colorization, translation, and deletion proceed as follows:
(1) Sample points $[…, x_i, …]$ on a ray (as usual in NeRF).
(2) For each point, we query the DFF. We can now use Eq. 5 to calculate the softmax probability of the coordinate being matched with a set of queries as $p \in [0, 1]$. With a single query, we may use cosine similarity (see paper line 252 or L.131). We now define "the coordinate is selected by the query" if $p$ is above a user-defined threshold, otherwise "not selected".
(3) If *not* selected, we calculate density $\sigma(x)$ and color $c(x)$ at $x$ via the vanilla NeRF.
(4) If selected, we may apply the following transforms:
(4-A) Deletion (Figure 4): We set the density $\sigma(x)$ of the point to zero.
(4-B) Color editing: We query the NeRF for density $\sigma(x)$ and color $c(x)$ by querying the NeRF. The color is then edited by a colorization function $b$, i.e., it is transformed to $b(c(x))$.
(4’-C) Translation / rescaling: Geometric transformation needs another step before performing (2). We first compute a deformed point coordinate $x’$: $x’$ is computed by applying the inverse of the editing transformation; that is, $x’ = g^{-1}(x)$. For translation, $g$ would be a simple addition with a vector. If $x’$ is selected by the query, $x’$ is used instead of $x$ for calculating color and density. If both $x$ and $x’$ are selected and have non-zero density (e.g., the boundary between the deformed apple and others in Figure 7), we mix their colors $c(x)$ and $c(x')$ in the ratio of their alphas at the point for simplicity.
(5) Finally, as usual, we perform volume rendering with the series of (density, color) tuples.

We may also use two different NeRF MLPs, and simple alpha-compositing to mix scenes as done in Figure 8.

Note that these edits do *not* require re-training of the NeRF. An exception is the CLIPNeRF-based editing (Fig. 8, line 323 onwards). Here, we perform an edit using the method defined in CLIPNeRF, which, however, fails to accurately select relevant 3D regions. We thus use DFF for selecting the relevant region, and then compose the background taken from the un-edited NeRF with the CLIPNeRF-edited foreground NeRF.


## Does each editing operation require re-training of the model?
cXkV inquires whether each editing operation requires re-training the model. Of all the edits we show, only the CLIPNeRF edits require fine-tuning of a NeRF by minimizing the embedding loss between a rendered image and the CLIP text embedding. **All other edits, i.e. decomposition and selection and subsequent re-scaling, scene composition, color change, deletion, etc., do not require re-training, and incur negligible computational cost.**


## Training Details
We agree that training details would benefit the clarity of the paper. We will integrate all training details (currently found in the supplement) in Section 5.


## Comparisons with other NeRF editing methods
We agree that a quantitative comparison with a baseline NeRF editing method would make the paper stronger. Alas, we could not identify any dataset or method that would enable us to perform such a quantitative comparison, as this would require pairs of ground-truth edited and unedited scenes, and such a dataset is, to the best of our knowledge, not available. We are grateful for any proposals of how such a quantitative comparison could be performed, and what baselines we should consider.

---

> ### Author Response · Authors · 2022-08-02
> **General Clarifications**
>
> ## Rigor of Claim
> Reviewer XqPY points out that our claim of distilling self-supervised pre-trained image feature extractors is not rigorous, as we also distill the LSeg feature extractor. We will re-formulate this claim to instead state “feature encoders pre-trained in both self-supervised and supervised frameworks”. We note that fundamentally, our framework does not make any assumptions on how the image encoder is trained, and we note that much of our results are accomplished with the DINO encoder, which is indeed trained fully self-supervised. We argue that the fact that our method is agnostic to the training method is a _strength_ of the proposed approach, as it lays the groundwork to “lift” any 2D image feature extractor - supervised or not - into 3D to enable editing and potentially other downstream tasks.
>
> ## Train Feature Branch Independently
> It is indeed the case that the feature and radiance fields can be trained independently, with completely separate MLPs. In line 240 and Appendix, Table 3, we report quantitative results of this approach and note that it performs comparably to a version where feature field and radiance field share part or all of the MLP.
> It is further indeed possible and an interesting direction for future work to not train the NeRF on RGB pictures at all, and use the proposed Feature Field by itself as a representation for downstream tasks in vision and robotics. While we don’t make any claims on 3D semantic segmentation performance, our analysis nevertheless indicates that the proposed approach could, in principle, be used to “lift” any predictions made by 2D image processing models into a consistent 3D representation, which is a promising direction for future work.
> Meanwhile, as feature maps may be 3D inconsistent and of lower frequency, this might come with additional challenges on the accuracy of reconstructed geometry - we leave such an investigation to future work.
> We will add this discussion to the experiments section, as well as the discussion of future work.
>
> ## Novelty
> We propose to distill a pre-trained 2D feature extractor into a 3D-structured neural scene representation, which, to the best of our knowledge, is a novel approach. The proposed approach achieves a high quality, and convincingly enables a down-stream task of significant interest, NeRF editing. With this work, we believe we motivate exciting future work towards distilling *any* pre-trained 2D image model into Neural Radiance Fields, which we believe to be an impactful and fascinating area of future work.
>
> ## Performance on Semantic Segmentation
> XqPY argues that the fact that we do not claim SOTA performance on 3D semantic segmentation weakens our contributions. They further argue that our paper “basically introduces a distillation method for NeRF semantic segmentation”.
>
> We would like to highlight that semantic segmentation is *not* the scope of this paper. The scope of our paper is to demonstrate a novel approach towards selection of 3D regions and subsequent editing of the underlying 3D representation, by distilling a pre-trained 2D feature extractor such as DINO. Indeed, the proposed approach is significantly more general than distilling a pre-trained semantic segmentation model, which would allow selection _only of areas that were in the set of classes that the model was pre-trained for_. Instead, we demonstrate distillation of both a pre-trained _self-supervised_ feature extractor, DINO, and of LSeg, a more application-specific feature extractor, enabling edits of uncommon objects, such as the dinosaur skeleton.
>
> We agree that 3D semantic segmentation is an interesting direction for future work, which is the reason for the inclusion of such a benchmark in the present paper. However, we believe that the set of experiments and results we show is absolutely necessary to properly evaluate our selection and editing claims. The analysis proposed by the reviewer, while indeed very valuable, would far exceed the scope of the present paper, which is already at the page limit.
>
> The reviewer further proposes to benchmark semantic segmentation approaches on other indoor scenes of the LLFF dataset. However, we note that LLFF neither features ground-truth semantic segmentation, nor are the LLFF images appropriately covered by the semantic classes of any existing dataset, such as SCANNET (please see the SCANNET classes [here](http://www.scan-net.org/ScanNet/Tasks/Benchmark/classes_SemVoxLabel-nyu40id.txt)). We thus believe a semantic segmentation benchmark on LLFF to be infeasible.
>
> Finally, we note that the [current state-of-the-art on SCANNET ](https://arxiv.org/pdf/2110.02210v2.pdf), (see leaderboard [here](https://paperswithcode.com/sota/semantic-segmentation-on-scannet)) indeed is based on the MinkowskiNet42 architecture that we used in our 3D segmentation benchmark, and thus, we believe that the paper in its present form serves as a motivation and starting point for future work in this direction.

---

> > ### Author Response · Authors · 2022-08-02
> > **General Clarifications**
> >
> > ## Distilled Feature Fields for other NeRF Variants
> > We agree that demonstrating DFF on other NeRF variants would serve to highlight the generality of the proposed approach. We note that our method does not make any assumptions on the parameterization of the NeRF - the proposed approach is agnostic to the parameterization and thus in principle compatible with voxelgrid-, tri-plane-, ground-plan and other types of scene representations. We can also use an independent MLP for feature fields that uses a parameterization of the feature field **different** from that of the feature field, as demonstrated in line 240 and Appendix, Table 3 (see also "Train Feature Branch Independently"-section in this response). We did not succeed in running such an experiment in time for the rebuttal, but will include such a result in the camera-ready paper. We will add this discussion to the methods section, as well as to the discussion section.
> >
> > ## Discussion of Limitations
> > While we discuss several failure modes of our approach (lines 286, 288, 317), we will compile them in a dedicated section “Limitations”. We will further add a discussion of the point raised by reviewer 1TRr: It is indeed the case that if the teacher network can only output low-resolution feature maps, the resulting supervision might be noisy and harm training fine-grained feature fields, although the DFF could have some denoising effects thanks to multi-view fusion. This is also discussed in [SemanticNeRF's](https://arxiv.org/pdf/2103.15875.pdf): the lower the resolution, the lower the segmentation performance. We will further merge in the discussion in appendix line 812, where we point out that the feature field inherits certain weaknesses of the teacher network.

---

### Meta-Review · Area_Chair_cyRv · 2022-08-26

**Recommendation:** Accept
**Confidence:** Certain

**Metareview:**

The paper proposes an approach for manipulating 3d scenes represented with implicit neural representations (NeRF-like), via distilling 2D feature extractors into a 3D feature field. The method shows convincing qualitative results on scene editing and promising quantitative results on semantic segmentation.

All reviewers are (to a different degree) positive about the paper, noting good presentation, interesting and fairly novel approach, fairly thorough and convincing results (it would be nice to have quantitative results on scene editing, but that's quite non-trivial).

Overall, this is an interesting and well-executed paper and I am happy to recommend acceptance.

**Award:**

No

---

### Decision · Program_Chairs · 2022-09-14

Accept